# HYPERPARAMETER OPTIMIZATION: A SPECTRAL APPROACH

**Elad Hazan**
Princeton University and Google Brain
ehazan@cs.princeton.edu

**Adam Klivans**
Department of Computer Science
University of Texas at Austin
klivans@cs.utexas.edu

**Yang Yuan**
Department of Computer Science
Cornell University
yangyuan@cs.cornell.edu

## ABSTRACT

We give a simple, fast algorithm for hyperparameter optimization inspired by techniques from the analysis of Boolean functions. We focus on the high-dimensional regime where the canonical example is training a neural network with a large number of hyperparameters. The algorithm — an iterative application of compressed sensing techniques for orthogonal polynomials — requires only uniform sampling of the hyperparameters and is thus easily parallelizable.

Experiments for training deep neural networks on Cifar-10 show that compared to state-of-the-art tools (e.g., Hyperband and Spearmint), our algorithm finds significantly improved solutions, in some cases better than what is attainable by hand-tuning. In terms of overall running time (i.e., time required to sample various settings of hyperparameters plus additional computation time), we are at least an order of magnitude faster than Hyperband and Bayesian Optimization. We also outperform Random Search $8\times$.

Our method is inspired by provably-efficient algorithms for learning decision trees using the discrete Fourier transform. We obtain improved sample-complexy bounds for learning decision trees while matching state-of-the-art bounds on running time (polynomial and quasipolynomial, respectively).

## 1 INTRODUCTION

Large scale machine learning and optimization systems usually involve a large number of free parameters for the user to fix according to their application. A timely example is the training of deep neural networks for a signal processing application: the ML specialist needs to decide on an architecture, depth of the network, choice of connectivity per layer (convolutional, fully-connected, etc.), choice of optimization algorithm and recursively choice of parameters inside the optimization library itself (learning rate, momentum, etc.).

Given a set of hyperparameters and their potential assignments, the naive practice is to search through the entire grid of parameter assignments and pick the one that performed the best, a.k.a. "grid search". As the number of hyperparameters increases, the number of possible assignments increases exponentially and a grid search becomes quickly infeasible. It is thus crucial to find a method for automatic tuning of these parameters.

This auto-tuning, or finding a good setting of these parameters, is now referred to as hyperparameter optimization (HPO), or simply automatic machine learning (auto-ML). For continuous hyperparameters, gradient descent is usually the method of choice (Maclaurin et al., 2015; Luketina et al., 2015; Fu et al., 2016). Discrete parameters, however, such as choice of architecture, number of layers, connectivity and so forth are significantly more challenging. More formally, let

$$f : \{-1, 1\}^n \mapsto [0, 1]$$

be a function mapping hyperparameter choices to test error of our model. That is, each dimension corresponds to a certain hyperparameter (number of layers, connectivity, etc.), and for simplicity of illustration we encode the choices for each parameter as binary numbers $\{-1, 1\}$. The goal of HPO is to approximate the minimizer $x^* = \arg\min_{x \in \{0,1\}^n} f(x)$ in the following setting:

1. Oracle model: evaluation of $f$ for a given choice of hyperparameters is assumed to be very expensive. Such is the case of training a given architecture of a huge dataset.

2. Parallelism is crucial: testing several model hyperparameters in parallel is entirely possible in cloud architecture, and dramatically reduces overall optimization time.

3. $f$ is structured.

The third point is very important since clearly HPO is information-theoretically hard and $2^n$ evaluations of the function are necessary in the worst case. Different works have considered exploiting one or more of the properties above. The approach of Bayesian optimization (Snoek et al., 2012) addresses the structure of $f$, and assumes that a useful prior distribution over the structure of $f$ is known in advance. Multi-armed bandit algorithms (Li et al., 2016), and Random Search (Bergstra & Bengio, 2012), exploit computational parallelism very well, but do not exploit any particular structure of $f$[1]. These approaches are surveyed in more detail later.

## 1.1 OUR CONTRIBUTION

In this paper we introduce a new *spectral* approach to hyperparameter optimization. Our main idea is to make assumptions on the structure of $f$ in the Fourier domain. Specifically we assume that $f$ can be approximated by a sparse and low degree polynomial in the Fourier basis. This means intuitively that it can be approximated well by a decision tree.

The implication of this assumption is that we can obtain a rigorous theoretical guarantee: approximate minimization of $f$ over the boolean hypercube with **function evaluations only linear in sparsity that can be carried out in parallel**. We further give improved heuristics on this basic construction and show experiments showing our assumptions are validated in practice for HPO as applied to deep learning over image datasets.

Thus our contributions can be listed as:

- A new spectral method called *Harmonica* that has provable guarantees: sample-efficient recovery if the underlying objective is a sparse (noisy) polynomial and easy to implement on parallel architectures.

- We demonstrate significant improvements in accuracy, sample complexity, and running time for deep neural net training experiments. We compare ourselves to state-of-the-art solvers from Bayesian optimization, Multi-armed bandit techniques, and Random Search. Projecting to even higher numbers of hyperparameters, we perform simulations that show several orders-of-magnitude of speedup versus Bayesian optimization techniques.

- Improved bounds on the sample complexity of learning noisy, size $s$ decision trees over $n$ variables under the uniform distribution. We observe that the classical sample complexity bound of $n^{O(\log(s/\varepsilon))}$ due to Linial et al. (1993) can be improved to quadratic in the size of the tree $\tilde{O}(s^2/\varepsilon \cdot \log n)$ while matching the best known quasipolynomial bound in running time.

## 1.2 PREVIOUS WORK

The literature on discrete-domain HPO can be roughly divided into two: probabilistic approaches and decision-theoretic methods. In critical applications, researchers usually use a grid search over all parameter space, but that becomes quickly prohibitive as the number of hyperparameter grows. Gradient-based methods such as (Maclaurin et al., 2015; Luketina et al., 2015; Fu et al., 2016; Bengio, 2000) are applicable only to continuous hyperparameters which we do not consider. Neural network structural search based on reinforcement learning is an active direction (Baker et al., 2016; Zoph & Le, 2016; Zhong et al., 2017), which usually needs many samples of network architectures.

---

[1] except that they could implicitly utilize smoothness or other local properties of the space.

**Probabilistic methods and Bayesian optimization.** Bayesian optimization (BO) algorithms (Bergstra et al., 2011; Snoek et al., 2012; Swersky et al., 2013; Snoek et al., 2014; Gardner et al., 2014; Wang et al., 2013; Ilievski et al., 2017) tune hyperparameters by assuming a prior distribution of the loss function, and then keep updating this prior distribution based on the new observations. Each new observation is selected according to an acquisition function, which balances exploration and exploitation such that the new observation gives us a better result, or helps gain more information. The BO approach is inherently serial and difficult to parallelize, and its theoretical guarantees have thus far been limited to statistical consistency (convergence in the limit).

**Decision-theoretic methods.** Perhaps the simplest approach to HPO is random sampling of different choices of parameters and picking the best amongst the chosen evaluations (Bergstra & Bengio, 2012). It is naturally very easy to implement and parallelize. Upon this simple technique, researchers have tried to allocate different budgets to the different evaluations, depending on their early performance. Using adaptive resource allocation techniques found in the multi-armed bandit literature, Successive Halving (SH) algorithm was introduced (Jamieson & Talwalkar, 2016). Hyperband further improves SH by automatically tuning the hyperparameters in SH (Li et al., 2016).

**Learning decision trees.** Prior work for learning decision trees (more generally Boolean functions that are approximated by low-degree polynomials) used the celebrated "low-degree" algorithm of Linial et al. (1993). Their algorithm uses random sampling to estimate each low-degree Fourier coefficient to high accuracy.

We make use of the approach of Stobbe & Krause (2012), who showed how to learn low-degree, sparse Boolean functions using tools from compressed sensing (similar approaches were taken by Kocaoglu et al. (2014) and Negahban & Shah (2012)). We observe that their approach can be extended to learn functions that are both "approximately sparse" (in the sense that the $L_1$ norm of the coefficients is bounded) and "approximately low-degree" (in the sense that most of the $L_2$ mass of the Fourier spectrum resides on monomials of low-degree). This implies the first decision tree learning algorithm with polynomial sample complexity that handles adversarial noise. In addition, we obtain the optimal dependence on the error parameter $\varepsilon$.

For the problem of learning *exactly* $k$-sparse Boolean functions over $n$ variables, Haviv & Regev (2015) have recently shown that $O(nk \log n)$ uniformly random samples suffice. Their result is not algorithmic but does provide an upper bound on the information-theoretic problem of how many samples are required to learn. The best algorithm in terms of running time for learning $k$-sparse Boolean functions is due to Feldman et al. (2009), and requires time $2^{\Omega(n/\log n)}$. It is based on the Blum et al. (2003) algorithm for learning parities with noise.

**Techniques.** Our methods are heavily based on known results from the analysis of boolean functions as well as compressed sensing.

## 2 SETUP AND DEFINITIONS

The problem of hyperparameter optimization is that of minimizing a discrete, real-valued function, which we denote by $f : \{-1,1\}^n \mapsto [-1,1]$ (we can handle arbitrary inputs, binary is chosen for simplicity of presentation).

In the context of hyperparameter optimization, function evaluation is very expensive, although parallelizable, as it corresponds to training a deep neural net. In contrast, any computation that does not involve function evaluation is considered less expensive, such as computations that require time $\Omega(n^d)$ for "somewhat large" $d$ or are subexponential (we still consider runtimes that are exponential in $n$ to be costly).

### 2.1 BASICS OF FOURIER ANALYSIS

The reader is referred to O'Donnell (2014) for an in depth treatment of Fourier analysis of Boolean functions. Let $f : \mathcal{X} \mapsto [-1,1]$ be a function over domain $\mathcal{X} \subseteq \mathbb{R}^n$. Let $\mathcal{D}$ a probability distribution on $\mathcal{X}$. We write $g \equiv_\varepsilon f$ and say that $f, g$ are $\varepsilon$**-close** if $\mathbb{E}_{x \sim \mathcal{D}}[(f(x) - g(x))^2] \leq \varepsilon$.

**Definition 1.** (Rauhut, 2010) We say a family of functions $\psi_1, \ldots, \psi_N$ ($\psi_i$ maps $\mathcal{X}$ to $\mathbb{R}$) is a *Random Orthonormal Family* with respect to $\mathcal{D}$ if

$$\mathbb{E}_{\mathcal{D}}[\psi_i(X) \cdot \psi_j(X)] = \delta_{ij} = \left\{ \begin{array}{ll} 1 & \text{if } i = j \\ 0 & \text{otherwise} \end{array} \right. .$$

The expectation is taken with respect to probability distribution $\mathcal{D}$. We say that the family is $K$-bounded if $\sup_{x \in \mathcal{X}} |\psi_i(x)| \leq K$ for every $i$. Henceforth we assume $K = 1$.

An important example of a random orthonormal family is the class of parity functions with respect to the uniform distribution on $\{-1, 1\}^n$:

**Definition 2.** A parity function on some subset of variables $S \subseteq [n]$ is the function $\chi_S : \{-1, 1\}^n \mapsto \{-1, 1\}$ where $\chi_S(x) = \prod_{i \in S} x_i$.

It is easy to see that the set of all $2^n$ parity functions $\{\chi_S\}$, one for each $S \subseteq [n]$, form a random orthonormal family with respect to the uniform distribution on $\{-1, 1\}^n$.

This random orthonormal family is often referred to as the Fourier basis, as it is a complete orthonormal basis for the class of Boolean functions with respect to the uniform distribution on $\{-1, 1\}^n$. More generally, for any $f : \{-1, 1\}^n \mapsto \mathbb{R}$, $f$ can be uniquely represented in this basis as $f(x) = \sum_{S \subseteq [n]} \hat{f}_S \chi_S(x)$ where $\hat{f}_S = \langle f, \chi_S \rangle = \mathbb{E}_{x \in \{-1, 1\}^n}[f(x)\chi_S(x)]$ is the Fourier coefficient corresponding to $S$ where $x$ is drawn uniformly from $\{-1, 1\}^n$. We also have Parseval's identity: $\mathbb{E}[f^2] = \sum_S \hat{f}_S^2$.

In this paper, we will work exclusively with the above parity basis. Our results apply more generally, however, to any orthogonal family of polynomials (and corresponding product measure on $\mathbb{R}^n$). For example, if we wished to work with continuous hyperparameters, we could work with families of Hermite orthogonal polynomials with respect to multivariate spherical *Gaussian* distributions.

We conclude with a definition of low-degree, approximately sparse (bounded $L_1$ norm) functions:

**Definition 3** (Approximately sparse function)**.** Let $\{\chi_S\}$ be the parity basis, and let $\mathcal{C}$ be a class of functions mapping $\{-1, 1\}^n$ to $\mathbb{R}$. Thus for $f \in \mathcal{C}$, $f = \sum_S \hat{f}(S)\chi_S$. We say a function $f \in C$ is $s$-**sparse** if $L_0(f) \leq s$, ie., f has at most $s$ nonzero entries in its polynomial expansion. $f$ is $(\varepsilon, d)$-**concentrated** if $\mathbb{E}[(f - \sum_{S, |S| \leq d} \hat{f}(S)\chi_S)^2] \geq 1 - \varepsilon$. $\mathcal{C}$ is $(\varepsilon, d, s)$-**bounded** if for every $f \in \mathcal{C}$, $f$ is $(\varepsilon, d)$-concentrated and in addition $\mathcal{C}$ has $L_1$ norm bounded by $s$, that is, for every $f \in \mathcal{C}$ we have $\sum_S |\hat{f}(S)| \leq s$.

It is easy to see that the class of functions with bounded $L_1$ norm is more general than sparse functions. For example, the Boolean AND function has $L_1$ norm bounded by 1 but is not sparse.

We also have the following simple fact:

**Fact 4.** *(Mansour, 1994) Let $f$ be such that $L_1(f) \leq s$. Then there exists $g$ such that $g$ is $s^2/\varepsilon$ sparse and $E[(f - g)^2] \leq \varepsilon$. The function $g$ is constructed by taking all coefficients of magnitude $\varepsilon/s$ or larger in $f$'s expansion as a polynomial.*

## 2.2 COMPRESSED SENSING AND SPARSE RECOVERY

In the problem of *sparse recovery*, a learner attempts to recover a sparse vector $x \in \mathbb{R}^n$ which is $s$ sparse, i.e. $\|x\|_0 \leq s$, from an observation vector $y \in R^m$ that is assumed to equal $y = Ax + e$, where $e$ is assumed to be zero-mean, usually Gaussian, noise. The seminal work of Candes et al. (2006); Donoho (2006) shows how $x$ can be recovered exactly under various conditions on the observation matrix $A \in \mathbb{R}^{m \times n}$ and the noise. The usual method for recovering the signal proceeds by solving a convex optimization problem consisting of $\ell_1$ minimization as follows (for some parameter $\lambda > 0$):

$$\min_{x \in \mathbb{R}^n} \left\{ \|x\|_1 + \lambda \|Ax - y\|_2^2 \right\}. \tag{1}$$

The above formulation comes in many equivalent forms (e.g., Lasso), where one of the objective parts may appear as a hard constraint.

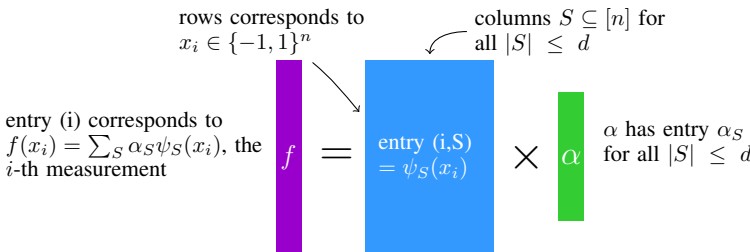

Figure 1: Compressed sensing over the Fourier domain: Harmonica recovers the Fourier coefficients of a sparse low degree polynomial $\sum_S \alpha_S \Psi_S(x_i)$ from observations $f(x_i)$ of randomly chosen points $x_i \in \{-1, 1\}^n$.

For our work, the most relevant extension of traditional sparse recovery is due to Rauhut (2010), who considers the problem of sparse recovery when the measurements are evaluated according to a *random orthonormal family*. More concretely, fix $x \in \mathbb{R}^n$ with $s$ non-zero entries. For $K$-bounded random orthonormal family $\mathcal{F} = \{\psi_1, \ldots, \psi_N\}$, and $m$ independent draws $z^1, \ldots, z^m$ from corresponding distribution $\mathcal{D}$ define the $m \times N$ matrix $A$ such that $A_{ij} = \psi_j(z^i)$. Rauhut gives the following result for recovering sparse vectors $x$:

**Theorem 5** (Sparse Recovery for Random Orthonormal Families, (Rauhut, 2010) Theorem 4.4)**.** *Given as input matrix $A \in \mathbb{R}^{m \times N}$ and vector $y$ with $y_i = Ax + e_i$ for some vector $e$ with $\|e\|_2 \le \eta\sqrt{m}$, mathematical program (1) finds a vector $x^*$ such that for constants $c_1$ and $c_2$, $\|x - x^*\|_2 \le c_1 \frac{\sigma_s(x)_1}{\sqrt{s}} + c_2\eta$ with probability $1 - \delta$ as long as for sufficiently large constant $C$, $m \ge CK^2 \log K \cdot s \log^3 s \cdot \log^2 N \cdot \log(1/\delta)$.*

The term $\sigma_s(x)_1$ is equal to $\min\{\|x - z\|_1, z \text{ is } s \text{ sparse}\}$. Recent work (Bourgain, 2014; Haviv & Regev, 2016) has improved the dependence on the polylog factors in the lower bound for $m$.

## 3  BASIC ALGORITHM AND MAIN THEORETICAL RESULTS

The main component of our spectral algorithm for hyperparameter optimization is given in Algorithm 1[2]. It is essentially an extension of sparse recovery (basis pursuit or Lasso) to the orthogonal basis of polynomials in addition to an optimization step. See Figure 1 for an illustration. We prove Harmonica's theoretical guarantee, and show how it gives rise to new theoretical results in learning from the uniform distribution.

In the next section we describe extensions of this basic algorithm to a more practical algorithm with various heuristics to improve its performance.

---

**Algorithm 1** Harmonica-1

---

1: Input: oracle for $f$, number of samples $T$, sparsity $s$, degree $d$, parameter $\lambda$.
2: Invoke PSR($f, T, s, d, \lambda$) (Procedure 2) to obtain $(g, J)$, where $g$ is a function defined on variables specified by index set $J \subseteq [n]$.
3: Set the variables in $[n] \setminus J$ to arbitrary values, compute a minimizer $x^\star \in \arg\min g(x)$.
4: **return** $x^\star$

---

**Theorem 6** (Noiseless recovery)**.** *Let $\{\psi_S\}$ be a 1-bounded orthonormal polynomial basis for distribution $\mathcal{D}$. Let $f : \mathbb{R}^n \mapsto \mathbb{R}$ be a $(0, d, s)$-bounded function as per definition 3 with respect to the basis $\psi_S$. Then Algorithm 1, in time $n^{O(d)}$ and sample complexity $T = \tilde{O}(s \cdot d \log n)$, returns $x^\star$ such that $x^\star \in \arg\min f(x)$.*

This theorem, and indeed most of the results in this paper, follows from the main recovery properties of Procedure 2. This recovery procedure satisfies the following main lemma. See its proof in Section A.1.

**Lemma 7** (Noisy recovery)**.** *Let $\{\psi_S\}$ be a 1-bounded orthonormal polynomial basis for distribution $\mathcal{D}$. Let $f : \mathbb{R}^n \mapsto \mathbb{R}$ be a $(\varepsilon/4, d, s)$-bounded as per definition 3 with respect to the*

---

[2]On line 3 of Algorithm 1, the minimizer can be found by enumerating all the possibilities, which is manageable if $g$ is a sparse low degree polynomial.

---

**Procedure 2** Polynomial Sparse Recovery (PSR)

1: Input: oracle for $f$, number of samples $T$, sparsity $s$, degree $d$, regularization parameter $\lambda$
2: Query $T$ random samples: $\{f(x_1), ...., f(x_T)\}$.
3: Solve sparse $d$-polynomial regression over all polynomials up to degree $d$

$$\arg\min_{\alpha \in \mathbb{R}^{\binom{n}{d}}} \left\{ \sum_{i=1}^{T} \left( \sum_{|S| \leq d} \alpha_S \psi_S(x_i) - f(x_i) \right)^2 + \lambda\|\alpha\|_1 \right\} \qquad (2)$$

4: Let $S_1, ..., S_s$ be the indices of the largest coefficients of $\vec{\alpha}$.
5: **return** $g \triangleq \sum_{i \in [s]} \alpha_{S_i} \psi_{S_i}(x)$ and $J = \cup_{i=1}^{s} S_i$

---

basis $\psi_S$. Then Procedure 2 finds a function $g \equiv_\varepsilon f$ in time $O(n^d)$ and sample complexity $T = \tilde{O}(s^2/\varepsilon \cdot d \log n)$.

**Remark:** Note that the above Lemma also holds in the *adversarial* or *agnostic* noise setting. That is, an adversary could add a noise vector $v$ to the labels received by the learner. In this case, the learner will see label vector $y = Ax + e + v$. If $\|v\|_2 \leq \sqrt{\gamma m}$, then we will recover a polynomial with squared-error at most $\varepsilon + O(\gamma)$ via re-scaling $\varepsilon$ by a constant factor and applying the triangle inequality to $\|e + v\|_2$.

While this noisy recovery lemma is the basis for our enhanced algorithm in the next section as well as the learning-theoretic result on learning of decision trees detailed in the next subsection, it does not imply recovery of the global optimum. The reason is that noisy recovery guarantees that we output a hypothesis *close* to the underlying function, but even a single noisy point can completely change the optimum.

Nevertheless, we can use our techniques to prove recovery of optimality for functions that are computed *exactly* by a sparse, low-degree polynomial (Theorem 6). See the proof in Section A.2.

### 3.1 APPLICATION: LEARNING DECISION TREES IN QUASI-POLYNOMIAL TIME AND POLYNOMIAL SAMPLE COMPLEXITY

Lemma 7 has important applications for learning (in the PAC model (Valiant, 1984)) well-studied function classes with respect to the uniform distribution on $\{-1, 1\}^n$[3]. For example, we obtain the first quasi-polynomial time algorithm for learning decision trees with respect to the uniform distribution on $\{-1, 1\}^n$ with *polynomial* sample complexity:

**Corollary 8.** *Let $\mathcal{X} = \{-1, 1\}^n$ and let $\mathcal{C}$ be the class of all decision trees of size $s$ on $n$ variables. Then $\mathcal{C}$ is learnable with respect to the uniform distribution in time $n^{O(\log(s/\varepsilon))}$ and sample complexity $m = \tilde{O}(s^2/\varepsilon \cdot \log n)$. Further, if the labels are corrupted by arbitrary noise vector $v$ such that $\|v\|_2 \leq \sqrt{\gamma m}$, then the output classifier will have squared-error at most $\varepsilon + O(\gamma)$.*

See the proof of Corollary 8 in Section A.3.

**Comparison with the "Low-Degree" Algorithm**. Prior work for learning decision trees (more generally Boolean functions that are approximated by low-degree polynomials) used the celebrated "low-degree" algorithm of Linial et al. (1993). Their algorithm uses random sampling to estimate each low-degree Fourier coefficient to high accuracy. In contrast, our approach is to use algorithms for compressed sensing to estimate the coefficients. Tools for compressed sensing take advantage of the incoherence of the design matrix and give improved results that seem unattainable from the "low-degree" algorithm.

For learning noiseless, Boolean decision trees, the low-degree algorithm uses quasipolynomial time and sample complexity $\tilde{O}(s^2/\varepsilon^2 \cdot \log n)$ to learn to accuracy $\varepsilon$. It is not clear, however, how to obtain any noise tolerance from their approach.

---

[3]All of our results also hold with respect to $\{0, 1\}^n$.

For general real-valued decision trees where $B$ is an upper bound on the maximum value at any leaf of a size $s$ tree, our algorithm will succeed with sample complexity $\tilde{O}(B^2 s^2 / \varepsilon \cdot \log n)$ and be tolerant to noise while the low-degree algorithm will use $\tilde{O}(B^4 s^2 / \varepsilon^2 \cdot \log n)$ (and will have no noise tolerance properties). Note our improvement in the dependence on $\varepsilon$ (even in the noiseless setting), which is a consequence of the RIP property of the random orthonormal family.

## 4 HARMONICA: THE FULL ALGORITHM

Rather than applying Algorithm 1 directly, we found that performance is greatly enhanced by iteratively using Procedure 2 to estimate the most influential hyperparameters and their optimal values.

In the rest of this section we describe this iterative heuristic, which essentially runs Algorithm 1 for multiple stages. More concretely, we continue to invoke the PSR subroutine until the search space becomes small enough for us to use a "base" hyperparameter optimizer (in our case either SH or Random Search).

The space of minimizing assignments to a multivariate polynomial is a highly non-convex set that may contain many distinct points. As such, we take an average of several of the best minimizers (of subsets of hyperparameters) during each stage.

In order to describe this formally we need the following definition of a restriction of function:

**Definition 9** (restriction (O'Donnell, 2014))**.** Let $f \in \{-1, 1\}^n \mapsto \mathbb{R}$, $J \subseteq [n]$, and $z \in \{-1, 1\}^J$ be given. We call $(J, z)$ a restriction pair of function $f$. We denote $f_{J,z}$ the function over $n - |J|$ variables given by setting the variables of $J$ to $z$.

We can now describe our main algorithm (Algorithm 3). Here $q$ is the number of stages for which we apply the PSR subroutine, and the restriction size $t$ serves as a tie-breaking rule for the best minimizers (which can be set to 1).

---

**Algorithm 3** Harmonica-$q$

1: Input: oracle for $f$, number of samples $T$, sparsity $s$, degree $d$, regularization parameter $\lambda$, number of stages $q$, restriction size $t$, base hyperparameter optimizer ALG.
2: **for** stage $i = 1$ to $q$ **do**
3:     Invoke PSR$(f, T, s, d, \lambda)$ (Procedure 2) to obtain $(g_i, J_i)$, where $g_i$ is a function defined on variables specified by index set $J_i \subseteq [n]$.
4:     Let $M_i = \{x_1^\star, ..., x_t^\star\} = \arg\min g_i(x)$ be the best $t$ minimizers of $g_i$.
5:     Let $f_i = \mathbb{E}_{k \in [t]}[f_{J_i, x_k^\star}]$ be the expected restriction of $f$ according to minimizers $M_i$.[4]
6:     Set $f = f_i$.
7: **end for**
8: **return** Search for the global minimizer of $f_q$ using base optimizer ALG

---

We defer the comparison of Harmonica and other algorithms in Section B.

## 5 EXPERIMENTS WITH TRAINING DEEP NETWORKS

We compare Harmonica[5] with Spearmint[6] (Snoek et al., 2012), Hyperband, SH[7] and Random Search. Both Spearmint and Hyperband are state-of-the-art algorithms, and it is observed that Random Search 2x (Random Search with doubled function evaluation resources) is a very competitive benchmark that beats many algorithms[8].

Our first experiment is over training residual network on Cifar-10 dataset[9]. We included 39 binary hyperparameters, including initialization, optimization method, learning rate schedule, momentum

---

[5]A python implementation of Harmonica can be found at https://github.com/callowbird/Harmonica

[6]https://github.com/HIPS/Spearmint.git

[7]We implemented a parallel version of Hyperband and SH in Lua.

[8]E.g., see (Recht, 2016a;b).

[9]https://github.com/facebook/fb.resnet.torch

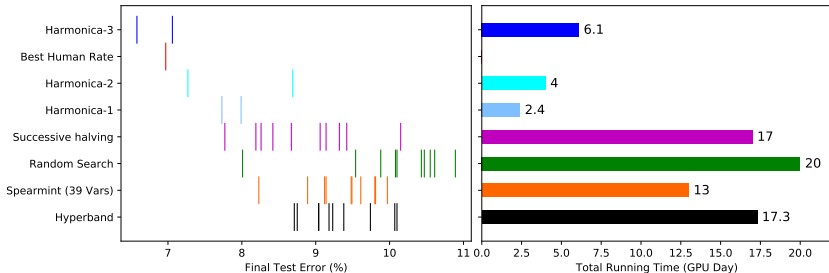

Figure 2: Distribution of the best results and running time of different algorithms

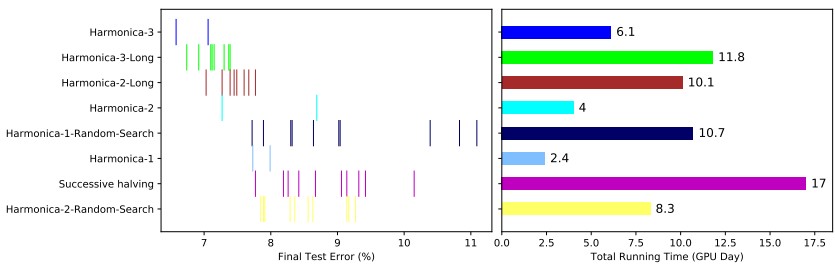

Figure 3: Comparing different variants of Harmonica with SH on test error and running time

rate, etc. Table 1 (Section C.1) details the hyperparameters considered. We also include 21 dummy variables to make the task more challenging. Notice that Hyperband, SH, and Random Search are agnostic to the dummy variables in the sense that they just set the value of dummy variables randomly, therefore select essentially the same set of configurations with or without the dummy variables. Only Harmonica and Spearmint are sensitive to the dummy variables as they try to learn the high dimensional function space. To make a fair comparison, we run Spearmint without any dummy variables.

As most hyperparameters have a consistent effect as the network becomes deeper, a common hand-tuning strategy is "tune on small network, then apply the knowledge to big network" (See discussion in Section C.3). Harmonica can also exploit this strategy as it selects important features stage-by-stage. More specifically, during the feature selection stages, we run Harmonica for tuning an 8 layer neural network with 30 training epochs. At each stage, we take 300 samples to extract 5 important features, and set restriction size $t = 4$ (see Procedure 2). After that, we fix all the important features, and run the SH or Random Search as our base algorithm on the big 56 layer neural network for training the whole 160 epochs[10]. To clarify, "stage" means the stages of the hyperparameter algorithms, while "epoch" means the epochs for training the neural network.

## 5.1 PERFORMANCE

We tried three versions of Harmonica for this experiment, Harmonica with 1 stage (Harmonica-1), 2 stages (Harmonica-2) and 3 stages (Harmonica-3). All of them use SH as the base algorithm. The top 10 test error results and running times of the different algorithms are depicted in Figure 2. SH based algorithms may return fewer than 10 results. For more runs of variants of Harmonica and its resulting test error, see Figure 3 (the results are similar to Figure 2).

**Test error and scalability:** Harmonica-1 uses less than $1/5$ time of Spearmint, $1/7$ time of Hyperband and $1/8$ time compared with Random Search, but gets better results than the competing algorithms. It beats the Random Search 8x benchmark (stronger than Random Search 2x benchmark of Li et al. (2016)). Harmonica-2 uses slightly more time, but is able to find better results.

---

[10]Other algorithms like Spearmint, Hyperband, etc. can be used as the base algorithms as well.

**Improving upon human-tuned parameters:** Harmonica-3 obtains a better test error (6.58%) as compared to the best hand-tuning rate 6.97% reported in (He et al., 2016)[11]. Harmonica-3 uses only 6.1 GPU days, which is less than half day in our environment, as we have 20 GPUs running in parallel. Notice that we did not cherry pick the results for Harmonica-3. In Section 5.3 we show by running Harmonica-3 for longer time, one can obtain a few other solutions better than hand tuning.

**Performance of provable methods:** Harmonica-1 has noiseless and noisy recovery guarantees (Lemma 7), which are validated experimentally.

## 5.2 AVERAGE TEST ERROR FOR EACH STAGE

We computed the average test error among 300 random samples for an 8 layer network with 30 epochs after each stage. See Figure 4 in Appendix. After selecting 5 features in stage 1, the average test error drops from 60.16 to 33.3, which indicates the top 5 features are very important. As we proceed to stage 3, the improvement on test error becomes less significant as the selected features at stage 3 have mild contributions.

## 5.3 HYPERPARAMETERS FOR HARMONICA

To be clear, Harmonica itself has six hyperparameters that one needs to set including the number of stages, $\ell_1$ regularizer for Lasso, the number of features selected per stage, base algorithm, small network configuration, and the number of samples per stage. Note, however, that we have reduced the search space of general hyperparameter optimization down to a set of only six hyperparameters. Empirically, our algorithm is robust to different settings of these parameters, and we did not even attempt to tune some of them (e.g., small network configuration).

**Base algorithm and #stages.** We tried different versions of Harmonica, including Harmonica with 1 stage, 2 stages and 3 stages using SH as the base algorithm (Harmonica-1, Harmonica-2, Harmonica-3), with 1 stage and 2 stages using Random Search as the base algorithm (Harmonica-1-Random-Search, Harmonica-2-Random-Search), and with 2 stages and 3 stages running SH as the base for longer time (Harmonica-2-Long, Harmonica-3-Long). As can be seen in Figure 3, most variants produce better results than SH and use less running time. Moreover, if we run SH for longer time, we will obtain more stable solutions with less variance in test error.

**Lasso parameters are stable.** See Table 3 in Appendix for stable range for regularization term $\lambda$ and the number of samples. Here stable range means as long as the parameters are set in this range, the top 5 features and the signs of their weights (which are what we need for computing $g(x)$ in Procedure 2) do not change. In other words, the feature selection outcome is not affected. When parameters are outside the stable ranges, usually the top features are still unchanged, and we miss only one or two out of the five features.

**On the degree of features.** We set degree to be three because it does not find any important features with degree larger than this. Since Lasso can be solved efficiently (less than 5 minutes in our experiments), the choice of degree can be decided automatically.

## 5.4 EXPERIMENTS WITH SYNTHETIC FUNCTIONS

Our second experiment considers a synthetic hierarchically bounded function $h(x)$. In this experiment, we showed that the optimization time of Harmonica is significantly faster than Spearmint, and the estimation error of Harmonica is linear in the noise level of the function. See Section C.4 for details.

## 6 ACKNOWLEDGEMENTS

We thank Sanjeev Arora for helpful discussions and encouragement. We thank anonymous reviewers for their helpful comments. Elad Hazan is supported by NSF grant 1523815. This project is supported by a Microsoft Azure research award and Amazon AWS research award.

---

[11] 6.97% is the rate obtained by residual network, and there are new network structures like wide residual network (Zagoruyko & Komodakis, 2016) or densenet (Huang et al., 2016) that achieve better rates for Cifar-10.

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

# A    MISSING PROOFS

## A.1    PROOF OF LEMMA 7

Recall the Chebyshev inequality:

**Fact 10** (Multidimensional Chebyshev inequality). *Let $X$ be an $m$ dimensional random vector, with expected value $\mu = \mathbb{E}[X]$, and covariance matrix $V = \mathbb{E}[(X - \mu)(X - \mu)^T]$.*

*If $V$ is a positive definite matrix, for any real number $\delta > 0$:*

$$\mathbb{P}(\sqrt{(X - \mu)^T V^{-1}(X - \mu)} > \delta) \leq \frac{m}{\delta^2}$$

For ease of notation we assume $K = 1$. Let $f$ be an $(\varepsilon/4, s, d)$-bounded function written in the orthonormal basis as $\sum_S \hat{f}(S)\psi_S$. We can equivalently write $f$ as $f = h + g$, where $h$ is a degree $d$ polynomial that only includes coefficients of magnitude at least $\varepsilon/4s$ and the constant term of the polynomial expansion of $f$.

Since $L_1(f) = \sum_S |\hat{f}_S| \leq s$, by Fact 4 we have that $h$ is $4s^2/\varepsilon + 1$ sparse. The function $g$ is thus the sum of the remaining $\hat{f}(S)\psi_S$ terms not included in $h$.

Draw $m$ (to be chosen later) random labeled examples $\{(z^1, y^1), \ldots, (z^m, y^m)\}$ and enumerate all $N = n^d$ basis functions $\psi_S$ for $|S| \leq d$ as $\{\psi_1, \ldots, \psi_N\}$. Form matrix $A$ such that $A_{ij} = \psi_j(z^i)$ and consider the problem of recovering $4s^2/\varepsilon + 1$ sparse $x$ given $Ax + e = y$ where $x$ is the vector of coefficients of $h$, the $i$th entry of $y$ equals $y^i$, and $e_i = g(z^i)$.

We will prove that with constant probability over the choice $m$ random examples, $\|e\|_2 \leq \sqrt{\varepsilon m}$. Applying Theorem 5 by setting $\eta = \sqrt{\varepsilon}$ and observing that $\sigma_{4s^2/\varepsilon+1}(x)_1 = 0$, we will recover $x'$ such that $\|x - x'\|_2^2 \leq c_2^2 \varepsilon$ for some constant $c_2$. As such, for the function $\tilde{f} = \sum_{i=1}^N x_i' \psi_i$ we will have $\mathbb{E}[\|h - \tilde{f}\|^2] \leq c_2^2 \varepsilon$ by Parseval's identity. Note, however, that we may rescale $\varepsilon$ by constant factor $1/(2c_2^2)$ to obtain error $\varepsilon/2$ and only incur an additional constant (multiplicative) factor in the sample complexity bound.

By the definition of $g$, we have

$$\|g\|^2 = \left( \sum_{S, |S|>d} \hat{f}(S)^2 + \sum_R \hat{f}(R)^2 \right) \tag{3}$$

where each $\hat{f}(R)$ is of magnitude at most $\varepsilon/4s$. By Fact 4 and Parseval's identity we have $\sum_R \hat{f}(R)^2 \leq \varepsilon/4$. Since $f$ is $(\varepsilon/4, d)$-concentrated we have $\sum_{S,|S|>d} \hat{f}(S)^2 \leq \varepsilon/4$. Thus, $\|g\|^2$ is at most $\varepsilon/2$. Therefore, by triangle inequality $\mathbb{E}[\|f - \tilde{f}\|^2] \leq \mathbb{E}[\|h - \tilde{f}\|^2] + \mathbb{E}[\|g\|^2] \leq \varepsilon$.

It remains to bound $\|e\|_2$. Note that since the examples are chosen independently, the entries $e_i = g(z^i)$ are independent random variables. Since $g$ is a linear combination of orthonormal monomials (not including the constant term), we have $\mathbb{E}_{z \sim D}[g(z)] = 0$. Here we can apply linearity of variance (the covariance of $\psi_i$ and $\psi_j$ is zero for all $i \neq j$) and calculate the variance

$$\mathbf{Var}(g(z^i)) = ( \sum_{S, |S|>d} \hat{f}(S)^2 + \sum_R \hat{f}(R)^2 )$$

With the same calculation as (3), we know $\mathbf{Var}(g(z^i))$ is at most $\varepsilon/2$.

Now consider the covariance matrix $V$ of the vector $e$ which equals $\mathbb{E}[ee^\top]$ (recall every entry of $e$ has mean 0). Then $V$ is a diagonal matrix (covariance between two independent samples is zero), and every diagonal entry is at most $\varepsilon/2$. Applying Fact 10 we have

$$\mathbb{P}(\|e\|_2 > \sqrt{\frac{\varepsilon}{2}}\delta) \leq \frac{m}{\delta^2}.$$

Setting $\delta = \sqrt{2m}$, we conclude that $\mathbb{P}(\|e\|_2 > \sqrt{\varepsilon m}) \leq \frac{1}{2}$. Hence with probability at least $1/2$, we have that $\|e\|_2 \leq \sqrt{\varepsilon m}$. From Theorem 5, we may choose $m = \tilde{O}(s^2/\varepsilon \cdot \log n^d)$. This completes the proof. Note that the probability $1/2$ above can be boosted to any constant probability with a constant factor loss in sample complexity.

## A.2 PROOF OF THEOREM 6

There are at most $N = n^d$ polynomials $\psi_S$ with $|S| \leq d$. Let the enumeration of these polynomials be $\psi_1, \ldots, \psi_N$. Draw $m$ labeled examples $\{(z^1, y^1), \ldots, (z^m, y^m)\}$ independently from $\mathcal{D}$ and construct an $m \times N$ matrix $A$ with $A_{ij} = \psi_j(z^i)$. Since $f$ can be written as an $s$ sparse linear combination of $\psi_1, \ldots, \psi_N$, there exists an $s$-sparse vector $x$ such that $Ax = y$ where the $i$th entry of $y$ is $y^i$. Hence we can apply Theorem 5 to recover $x$ exactly. These are the $s$ non-zero coefficients of $f$'s expansion in terms of $\{\psi_S\}$. Since $f$ is recovered exactly, its minimizer is found in the optimization step.

## A.3 PROOF OF COROLLARY 8

As mentioned earlier, the orthonormal polynomial basis for the class of Boolean functions with respect to the uniform distribution on $\{-1, 1\}^n$ is the class of parity functions $\{\chi_S\}$ for $S \subseteq \{-1, 1\}^n$. Further, it is easy to show that for Boolean function $f$, if $\mathbb{E}[(h - f)^2] \leq \varepsilon$ then $\mathbb{P}[\text{sign}(h(x)) \neq f(x)] \leq \varepsilon$. The corollary now follows by applying Lemma 7 and two known structural facts about decision trees: 1) a tree of size $s$ is $(\varepsilon, \log(s/\varepsilon))$-concentrated and has $L_1$ norm bounded by $s$ (see e.g., Mansour Mansour (1994)) and 2) by Fact 4, for any function $f$ with $L_1$ norm bounded by $s$ (i.e., a decision tree of size $s$), there exists an $s^2/\varepsilon$ sparse function $g$ such that $\mathbb{E}[(f - g)^2] \leq \varepsilon$. The noise tolerance property follows immediately from the remark after the proof of Lemma 7.

## B ALGORITHM ATTRIBUTES AND HEURISTICS

**Scalability.** If the hidden function if $s$-sparse, Harmonica can find such a sparse function using $\tilde{O}(s \log s)$ samples. If at every stage of Harmonica, the target function can be approximated by an $s$ sparse function, we only need $\tilde{O}(qs \log s)$ samples where $q$ is the number of stages. For real world applications such as deep neural network hyperparameter tuning, it seems (empirically) reasonable to assume that the hidden function is indeed sparse at every stage (see Section 5).

For Hyperband (Li et al., 2016), SH (Jamieson & Talwalkar, 2016) or Random Search, even if the function is $s$-sparse, in order to cover the optimal configuration by random sampling, we need $\Omega(2^s)$ samples.

**Optimization time.** Harmonica runs the Lasso (Tibshirani, 1996) algorithm after each stage to solve (2), which is a well studied convex optimization problem and has very fast implementations. Hyperband and SH are also efficient in terms of running time as a function of the number of function evaluations, and require sorting or other simple computations. The running time of Bayesian optimization is cubic in number of function evaluations, which limits applicability for large number of evaluations / high dimensionality, as we shall see in Section C.4.

**Parallelizability.** Harmonica, similar to Hyperband, SH, and Random Search, has straightforward parallel implementations. In every stage of those algorithms, we could simply evaluate the objective functions over randomly chosen points in parallel.

It is hard to run Bayesian optimization algorithm in parallel due to its inherent serial nature. Previous works explored variants in which multiple points are evaluated at the same time in parallel (Wu & Frazier, 2016), though speed ups do not grow linearly in the number of machines, and the batch size is usually limited to a small number.

**Feature Extraction.** Harmonica is able to extract important features with weights in each stages, which automatically sorts all the features according to their importance. See Section C.2.

---

[11] In order to evaluate $f_i$, we first sample $k \in [t]$ to obtain $f_{J_i, x_k^*}$, and then evaluate $f_{J_i, x_k^*}$.

## C  Experimental details

### C.1  Options

Table 1: 60 options used in Section 5

| Option Name | Description |
|---|---|
| 01. Weight initialization | Use standard initializations or other initializations? |
| 02. Weight initialization (Detail 1) | Xavier Glorot (Glorot & Bengio, 2010), Kaiming (He et al., 2015), $1/n$, or $1/n^2$? |
| 03. Optimization method | SGD or ADAM? (Kingma & Ba, 2014) |
| 04. Initial learning rate | $\geq 0.01$ or $< 0.01$? |
| 05. Initial learning rate (Detail 1) | $\geq 0.1$, $< 0.1$, $\geq 0.001$, or $< 0.001$? |
| 06. Initial learning rate (Detail 2) | 0.3, 0.1, 0.03, 0.01, 0.003, 0.001, 0.0003, or 0.0001? |
| 07. Learning rate drop | Do we need to decrease learning rate as we train? Yes or No? |
| 08. Learning rate first drop time | If drop learning rate, when is the first time to drop by $1/10$? Epoch 40 or Epoch 60? |
| 09. Learning rate second drop time | If drop learning rate, when is the second time to drop by $1/100$? Epoch 80 or Epoch 100? |
| 10.  Use momentum (Sutskever et al., 2013) | Yes or No? |
| 11. Momentum rate | If use momentum, rate is 0.9 or 0.99? |
| 12. Initial residual link weight | What is the initial residual link weight? All constant 1 or a random number in $[0,1]$? |
| 13. Tune residual link weight | Do we want to use back propagation to tune the weight of residual links? Yes or No? |
| 14. Tune time of residual link weight | When do we start to tune residual link weight? At the first epoch or epoch 10? |
| 15. Resblock first activation | Do we want to add activation layer after the first convolution? Yes or No? |
| 16. Resblock second activation | Do we want to add activation layer after the second convolution? Yes or No? |
| 17. Resblock third activation | Do we want to add activation layer after adding the residual link? Yes or No? |
| 18. Convolution bias | Do we want to have bias term in convolutional layers? Yes or No? |
| 19. Activation | What kind of activations do we use? ReLU or others? |
| 20. Activation (Detail 1) | ReLU, ReLU, Sigmoid, or Tanh? |
| 21. Use dropout (Srivastava et al., 2014) | Yes or No? |
| 22. Dropout rate | If use dropout, rate is high or low? |
| 23. Dropout rate (Detail 1) | If use dropout, the rate is 0.3, 0.2, 0.1, or 0.05? |
| 24. Batch norm (Ioffe & Szegedy, 2015) | Do we use batch norm? Yes or No? |
| 25. Batch norm tuning | If we use batch norm, do we tune the parameters in the batch norm layers? Yes or No? |
| 26. Resnet shortcut type | What kind of resnet shortcut type do we use? Identity or others? |
| 27. Resnet shortcut type (Detail 1) | Identity, Identity, Type B or Type C? |
| 28. Weight decay | Do we use weight decay during the training? Yes or No? |
| 29. Weight decay parameter | If use weight decay, what is the parameter? $1e-3$ or $1e-4$? |
| 30. Batch Size | What is the batch size we should use? Big or Small? |
| 31. Batch Size (Detail 1) | 256, 128, 64, or 32? |
| 32. Optnet | An option specific to the code[12]. Yes or No? |
| 33. Share gradInput | An option specific to the code. Yes or No? |
| 34. Backend | What kind of backend shall we use? cudnn or cunn? |
| 35. cudnn running state | If use cudnn, shall we use fastest of other states? |
| 36. cudnn running state (Detail 1) | Fastest, Fastest, default, deterministic |
| 37. nthreads | How many threads shall we use? Many or few? |

---

[12]https://github.com/facebook/fb.resnet.torch

| 38. nthreads (Detail 1) | 8, 4, 2, or 1? |
| 39-60. Dummy variables | Just dummy variables, no effect at all. |

See Table 1 for the specific hyperparameter options that we use in Section 5. For those variables with $k$ options ($k > 2$), we use $\log k$ binary variables under the same name to represent them. For example, we have two variables (01, 02) and their binary representation to denote four kinds of possible initializations: Xavier Glorot (Glorot & Bengio, 2010), Kaiming (He et al., 2015), $1/n$, or $1/n^2$.

## C.2 IMPORTANCE FEATURES

We show the selected important features and their weights during the first 3 stages in Table 2, where each feature is a monomial of variables with degree at most 3. We do not include the 4th stage because in that stage there are no features with nonzero weights.

**Smart choices on important options**. Based on Table 2, Harmonica will fix the following variables (sorted according to their importance): Batch Norm (Yes), Activation (ReLU), Initial learning rate ([0.001, 0.1]), Optimization method (Adam), Use momentum (Yes), Resblock first activation (Yes), Resblcok third activation (No), Weight decay (No if initial learning rate is comparatively small and Yes otherwise), Batch norm tuning (Yes). Most of these choices match what people are doing in practice.

**A metric for the importance of variables**. The features that Harmonica finds can serve as a metric for measuring the importance of different variables. For example, Batch Norm turns out to be the most significant variable, and ReLU is second important. By contrast, Dropout, when Batch Norm is presented, does not have significant contributions. This actually matches with the observations in (Ioffe & Szegedy, 2015).

**No dummy/irrelevant variables selected**. Although there are 21/60 dummy variables, we never select any of them. Moreover, the irrelevant variables like cudnn, backend, nthreads, which do not affect the test error, were not selected.

Table 2: Important features

| Stage | Feature Name | Weights |
|-------|-------------|---------|
| 1-1 | 24. Batch norm | 8.05 |
| 1-2 | 19. Activation | 3.47 |
| 1-3 | 04. Initial learning rate * 05. Initial learning rate (Detail 1) | 3.12 |
| 1-4 | 19. Activation * 24. Batch norm | -2.55 |
| 1-5 | 04. Initial learning rate | -2.34 |
| 1-6 | 28. Weight decay | -1.90 |
| 1-7 | 24. Batch norm * 28. Weight decay | 1.79 |
| 1-8 | 34. Optnet * 35. Share gradInput * 52. Dummy [13] | 1.54 |
| 2-1 | 03. Optimization method | -4.22 |
| 2-2 | 03. Optimization method * 10. Use momentum | -3.02 |
| 2-3 | 15. Resblock first activation | 2.80 |
| 2-4 | 10. Use momentum | 2.19 |
| 2-5 | 15. Resblock first activation * 17. Resblock third activation | 1.68 |
| 2-6 | 01. Good initialization | -1.26 |
| 2-7 | 01. Good initialization * 10. Use momentum | -1.12 |
| 2-8 | 01. Good initialization * 03. Optimization method | 0.67 |
| 3-1 | 29. Weight decay parameter | -0.49 |
| 3-2 | 28. Weight decay | -0.26 |
| 3-3 | 06. Initial learning rate (Detail 3) * 28. Weight decay | 0.23 |
| 3-4 | 25. Batch norm tuning | 0.21 |
| 3-5 | 28. Weight decay * 29. Weight decay parameter | 0.20 |

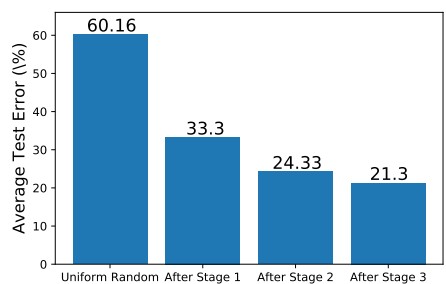

Figure 4: Average test error drops.

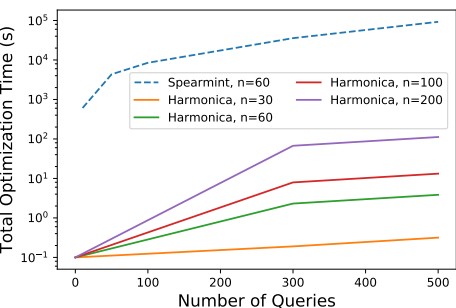

Figure 5: Optimization time comparison

Table 3: Stable ranges for parameters in Lasso

| Parameter | Stage 1 | Stage 2 | Stage 3 |
|---|---|---|---|
| $\lambda$ | $[0.01, 4.5]$ | $[0.1, 2.5]$ | $[0.5, 1.1]$ |
| #Samples | $\geq 250$ | $\geq 180$ | $\geq 150$ |

## C.3 GENERALIZING FROM SMALL NETWORKS TO BIG NETWORKS

In our experiments, Harmonica first runs on a small network to extract important features and then uses these features to do fine tuning on a big network. Since Harmonica finds significantly better solutions, it is natural to ask whether other algorithms can also exploit this strategy to improve performance.

Unfortunately, it seems that all the other algorithms do not naturally support feature extraction from a small network. For Bayesian Optimization techniques, small networks and large networks have different optimization spaces. Therefore without some modification, Spearmint cannot use information from the small network to update the prior distribution for the large network.

Random-search-based techniques are able to find configurations with low test error on the small network, which might be good candidates for the large network. However, based on our simulation, good configurations of hyperparameters from random search do not generalize from small networks to large networks. This is in contrast to important features in our (Fourier) space, which do seem to generalize.

To test the latter observation using Cifar-10 dataset, we first spent 7 GPU days on 8 layer network to find top 10 configurations among 300 random selected configurations. Then we apply these 10 configurations, as well as 90 locally perturbed configurations (each of them is obtained by switching one random option from one top-10 configuration), so in total 100 "promising" configurations, to the large 56 layer network. This simulation takes 27 GPU days, but the best test error we obtained is only 11.1%, even worse than purely random search. Since Hyperband is essentially a fast version of Random Search, it also does not support feature extraction.

Hence, being able to extract important features from small networks seems empirically to be a unique feature of Harmonica.

## C.4 EXPERIMENTS WITH SYNTHETIC FUNCTIONS

Our second experiment considers a synthetic hierarchically bounded function $h(x)$. We run Harmonica with 100 samples, 5 features selected per stage, for 3 stages, using degree 3 features. See Figure 5 for optimization time comparison. We only plot the optimization time for Spearmint when $n = 60$, which takes more than one day for 500 samples. Harmonica is several magnitudes faster than Spearmint. In Figure 6, we show that Harmonica is able to estimate the hidden function with error proportional to the noise level.

---

[13]This is an interesting feature. In the code repository that we use, optnet, shared gradInput are two special options of the code and cannot be set true at the same time, otherwise the training becomes unpredictable.

The synthetic function $h(x) \in \{-1, +1\}^n \to \mathbb{R}$ is defined as follows. $h(x)$ has three stages, and in $i$-th stage ($i = 0, 1, 2$), it has $32^i$ sparse vectors $s_{i,j}$ for $j = 0, \cdots, 32^i - 1$. Each $s_{i,j}$ contains 5 pairs of weight $w_{i,j}^k$ and feature $f_{i,j}^k$ for $k = 1, \cdots 5$, where $w_{i,j}^k \in [10 + 10^{-i}, 10 + 10^{2-i}]$. and $f_{i,j}^k$ is a monomial on $x$ with degree at most 3. Therefore, for input $x \in \mathbb{R}^n$, the sparse vector $s_{i,j}(x) = \sum_{k=1}^5 w_{i,j}^k f_{i,j}^k(x)$. Since $x \in \{-1, +1\}^n$, $f_{i,j}^k(x)$ is binary. Therefore, $\{f_{i,j}^k(x)\}_{k=1}^5$ contains 5 binaries and represents a integer in $[0, 31]$, denoted as $c_{i,j}(x)$. Let $h(x) = s_{1,1}(x) + s_{2,c_{1,1}(x)}(x) + s_{3,c_{1,1}(x)*32+c_{2,c_{1,1}(x)}(x)}(x) + \xi$, where $\xi$ is the noise uniformly sampled from $[-A, A]$ ($A$ is the noise level). In other words, in every stage $i$ we will get a sparse vector $s_{i,j}$. Based on $s_{i,j}(x)$, we pick a the next sparse function and proceed to the next stage.

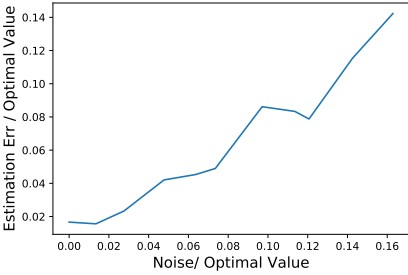

Figure 6: The estimation error of Harmonica is linear in noise level.

