# OpenReview forum: "Hyperparameter optimization: a spectral approach"
_ICLR.cc/2018/Conference — Accept (Poster)_

### Official Review · AnonReviewer3 · 2017-11-28
**The paper contains a strong theoretical result that is a bit out of context with the main theme of the paper. The algorithm presented shows promising results for optimizing hyperparameters when the number of hyperparameters > 6.**

**Rating:** 6
**Confidence:** 4

**Review:**

This paper looks at the problem of optimizing hyperparameters under the assumption that the unknown function can be approximated by a sparse and low degree polynomial in the Fourier basis. The main result is that the approximate minimization can be performed over the boolean hypercube where the number of evaluations is linear in the sparsity parameter.

In the presented experiments, the new spectral method outperforms the tool based on the Bayesian optimization, technique based on MAB and random search. Their result also has an application in learning decision trees where it significantly improves the sample complexity bound.

The main theoretical result, i.e., the improvement in the sample complexity when learning decision trees, looks very strong. However, I find this result to be out of the context with the main theme of the paper.

I find it highly unlikely that a person interested in using Harmonica to find the right hyperparamters for her deep network would also be interested in provable learning of decision trees in quasi-polynomial time along with a polynomial sample complexity. Also the theoretical results are developed for Harmonica-1 while Harmonica-q is the main method used in the experiments.

When it comes to the experiments only one real-world experiment is present. It is hard to conclude which method is better based on a single real-world experiment. Moreover, the plots are not very intuitive, i.e., one would expect that Random Search takes the smallest amount of time. I guess the authors are plotting the running time that also includes the time needed to evaluate different configurations. If this is the case, some configurations could easily require more time to evaluate than the others. It would be useful to plot the total number of function evaluations for each of the methods next to the presented plots.

It is not clear what is the stopping criterion for each of the methods used in the experiments. One weakness of Harmonica is that it has 6 hyperparameters itself to be tuned. It would be great to see how Harmonica compares with some of the High-dimensional Bayesian optimization methods.

Few more questions:

Which problem does Harmonica-q solves that is present in Harmonica-1, and what is the intuition behind the fact that it achieves better empirical results?

How do you find best t minimizers of g_i in line 4 of Algorithm 3?

---

> ### Author Response · Authors · 2017-12-12
> **Reply**
>
> Thank you for your summary and comments!
>
> 1. A practitioner can certainly safely ignore the theory and use Harmonica in practice.  The practitioner may, however, be interested to know that our approach is principled and comes with some provable guarantees (or may simply wonder where the approach came from).  As such, we think describing the relationship to decision-tree learning is valuable.
>
> 2. Re experiments: we find the CIFAR10 dataset to be a challenging one and representative of training deep neural networks (certainly it is the most intensely studied).  Finding settings better than hand-tuning indicates promise with our general approach.  There is always room for much more experimentation. We hope others will build on this new approach.
>
> 3. stopping criterion:  we allowed all algorithms (except Harmonica) to run for at least 17 days. Spearmint did not finish before the submission deadline (we will update the results). Subsequently, we have found it to run 2-3x slower than Harmonica and produce worse hyperparameter settings.
>
> 4. Harmonica-q vs Harmonica-1:  “q” is a heuristic that gives improved performance in practice.  The intuition is that there are few important variables in the objective, and after fixing them to the optimal value, this holds recursively - again there are few important variables (of 2nd order), and so on…
>
> 5. minimizers of g_i in line 4 of Algorithm 3:   this is a great question.
> We do it by enumerating all the possibilities. Here is where the assumptions come in: since we have a k-sparse degree-d polynomial, there are only 2^{k d} options, and this is manageable in this setting.

---

### Official Review · AnonReviewer2 · 2017-11-29
**Interesting theoretical ideas, but unclear how practical the proposed approach really is**

**Rating:** 6
**Confidence:** 3

**Review:**

- algorithm 1 has a lot of problem specific hyperparametes that may be difficult to get right. Not clear how important they are
- they analyze the simpler (analytically and likely computationally) Boolean hyperparameter case (each hyperparameter is binary). Not a realistic setting. In their experiments they use these binary parameter spaces so I'm not sure how much I buy that it is straightforward to use continuous valued polynomials.
- interesting idea but I think it's more theoretical than practical. Feels like a hammer in need of a nail.

---

> ### Author Response · Authors · 2017-12-12
> **Reply**
>
> Thank you for your comments!
>
> 1. Continuous vs. Boolean:   the Boolean setting is actually without loss of generality because we can search over a continuous range via binary search on discrete variables.  This seems to work well in practice.
>
> 2. Our theory works for any domain; the discrete non-Boolean case is handled just the same.
>
> 3. The number of hyperparameters for Harmonica is significantly lower than the input number (6 vs. 60, or any input #) and manageable by grid search.  We have also found that Harmonica is stable with respect to these six hyperparameters.
>
> 4. “hammer in need of a nail” suggests a complicated algorithm. This is far from the truth - the algorithm is very simple - just run LASSO on uniformly sampled measurements over the Fourier representation of the objective (and recurse). This is arguably simpler than Bayesian Optimization, or reinforcement learning based approaches, which require sophisticated updating and handling of prior distributions.

---

### Official Review · AnonReviewer1 · 2017-11-30
**Good paper on hyperparameter optimization using techniques from compressed sensing**

**Rating:** 9
**Confidence:** 5

**Review:**

The paper is about hyperparameter optimization, which is an important problem in deep learning due to the large number of hyperparameters in contemporary model architectures and optimization algorithms.

At a high-level, hyperparameter optimization (for the challenging case of discrete variables) can be seen as a black-box optimization problem where we have only access to a function evaluation oracle (but no gradients etc.). In the entirely unstructured case, there are strong lower bounds with an exponential dependence on the number of hyperparameters. In order to sidestep these impossibility results, the current paper assumes structure in the unknown function mapping hyperparameters to classification accuracy. In particular, the authors assume that the function admits a representation as a sparse and low-degree polynomial. While the authors do not empirically validate whether this is a good model of the unknown function, it appears to be a reasonable assumption (the authors *do* empirically validate their overall approach).

Based on the sparse and low-degree assumption, the paper introduces a new algorithm (called Harmonica) for hyperparameter optimization. The main idea is to leverage results from compressed sensing in order to recover the sparse and low-degree function from a small number of measurements (i.e., function evaluations). The authors derive relevant sample complexity results for their approach. Moreover, the method also yields new algorithms for learning decision trees.

In addition to the theoretical results , the authors conduct a detailed study of their algorithm on CIFAR10. They compare to relevant recent work in hyperparameter optimization (Bayesian optimization, random search, bandit algorithms) and find that their method significantly improves over prior work. The best parameters found by Harmonica improve over the hand-tuned results for their "base architecture" (ResNets).

Overall, I find the main idea of the paper very interesting and well executed, both on the theoretical and empirical side. Hence I strongly recommend accepting this paper.


Small comments and questions:

1. It would be interesting to see how close the hyperparameter function is to a low-degree and sparse polynomial (e.g., MSE of the best fit).

2. A comparison without dummy parameters would be interesting to investigate the performance differences between the algorithms in a lower-dimensional problem.

3. The current paper does not mention the related work on hyperparameter optimization using reinforcement learning techniques (e.g., Zoph & Le, ICLR 2017). While it might be hard to compare to this approach directly in experiments, it would still be good to mention this work and discuss how it relates to the current paper.

4. Did the authors tune the hyperparameters directly using the CIFAR10 test accuracy? Would it make sense to use a slightly smaller training set and to hold out say 5k images for hyperparameter evaluation before making the final accuracy evaluation on the test set? The current approach could be prone to overfitting.

5. While random search does not explicitly exploit any structure in the unknown function, it can still implicitly utilize smoothness or other benign properties of the hyperparameter space. It might be worth adding this in the discussion of the related work.

6. Algorithm 1: Why is the argmin for g_i  (what does the index i refer to)?

7. Why does PSR truncate the indices in alpha? At least in "standard" compressed sensing, the Lasso also has recovery guarantees without truncation (and empirically works sometimes better without).

9. Definition 3: Should C be a class of functions mapping {-1, 1}^n to R?  (Note the superscript.)

10. On Page 3 we assume that K = 1, but Theorem 6 still maintains a dependence on K. It might be cleaner to either treat the general K case throughout, or state the theorem for K = 1.

11. On CIFAR10, the best hyperparameters do not improve over the state of the art with other models (e.g., a wide ResNet). It could be interesting to run Harmonica in the regime where it might improve over the best known models for CIFAR10.

12. Similarly, it would be interesting to see whether the hyperparameters identified by Harmonica carry over to give better performance on ImageNet. The authors claim in C.3 that the hyperparameters identified by Harmonica generalize from small networks to large networks. Testing whether the hyperparameters also generalize from a smaller to a larger dataset would be relevant as well.

---

> ### Author Response · Authors · 2017-12-12
> **Reply**
>
> Thank you for your summary and comments! Answers to your questions:
>
> 1. Great suggestion. We have shown that the function does fit a low-degree polynomial by merit of optimization, but an MSE test is a good idea, and we’ll do that.
>
> 2. In fact, besides Spearmint (which runs without dummy parameters in our experiment) and Harmonica, other algorithms like random search, successive halving or hyperband will have exactly the same performance with/without dummy variables as they are based on random search in the parameter space. Therefore, by removing the dummy variables, only Harmonica might give better performance while the others will stay the same.
> So in short, the experiment is *non-favorable* to Harmonica with respect to dummy variables, showing its robustness.
>
> 3. Certainly, we will add discussion about this paper. The difficulty from comparing comes from the fact that the RL approach is inherently sequential, needing more information to proceed. Our approach is also based on a different assumption (sparse low degree polynomial).
>
> 4. We did not try this because our main goal was to try to do an apples to apples comparison of hyperparameter settings found by other algorithms on the entire training set (and indeed we found some that are even better than hand-tuning as in Figure 2).
>
> 5. Right, we’ll add discussion.
>
> 6. Typo, thanks!
>
> 7. This is simply a heuristic we tried, but it is definitely worth investigating no truncation (which we didn’t investigate enough).
>
> 9. Typo, yes!
>
> 10. Yes, we will remove the Ks.
>
> 11. When we started the project (around Sep, 2016), resnet was considered to be a pretty good model, but now maybe densenet is better.  Note that in Resnet, Harmonica does do better than best hand-tuned model, we hope same is true for densenet.
>
> 12. We did not have enough resources to do hyperparameter tuning for Imagenet, but intend to try this idea for CIFAR100 with densenet (i.e., using a subset of the data first).

---

### Decision · Program_Chairs · 2018-01-29
**ICLR 2018 Conference Acceptance Decision**

**Decision:**

Accept (Poster)

**Comment:**

This paper introduces an algorithm for optimization of discrete hyperparameters based on compressed sensing, and compares against standard gradient-free optimization approaches.

As the reviewers point out, the provable guarantees (as is usually the case) don't quite make it to the main results section, but are still refreshing to see in hyperparameter optimization.

The method itself is relatively simple compared to full-featured Bayesopt (spearmint), although not as widely applicable.